# Against the Norm: Do Not Rely on Serum C-Reactive Protein and White Blood Cell Count Only When Assessing Eradication of Periprosthetic Joint Infection

**DOI:** 10.3390/antibiotics11091174

**Published:** 2022-08-31

**Authors:** Farouk Khury, Moritz Oltmanns, Michael Fuchs, Janina Leiprecht, Heiko Reichel, Martin Faschingbauer

**Affiliations:** 1Department of Orthopedic Surgery, University of Ulm, Oberer Eselsberg 45, 89081 Ulm, Germany; 2Division of Orthopedic Surgery, Rambam Medical Center, The Ruth and Bruce Rappaport Faculty of Medicine, HaAliya HaShniya St 8, Haifa 3109601, Israel

**Keywords:** periprosthetic joint infection, diagnostic criteria, CRP, WBC

## Abstract

Introduction: Periprosthetic joint infections (PJI) following primary arthroplasty continue to be a serious complication, despite advances in diagnostics and treatment. Two-stage revision arthroplasty has been commonly used as the gold standard for the treatment of PJI. However, much discussion persists regarding the interim of the two-stage procedure and the optimal timing of reimplantation. Serology markers have been proposed as defining parameters for a successful reimplantation. The objective of this matched-pair analysis was to assess the role of serum C-reactive protein (CRP) and white blood cell count (WBC) in determining infection eradication and proper timing of reimplantation. We investigated the delta (∆) change in CRP and WBC values prior to both stages of two-stage revision arthroplasty as a useful marker of infection eradication. Methods: We analyzed 39 patients and 39 controls, matched by propensity score matching (BMI, age, ASA-classification), with a minimum follow-up of 24 months and treated with a two-stage revision THA or TKA in our institution. Data of serum CRP and WBC values were gathered at two selected time points: prior to the explantation of the implant (preexplantation) and following the completion of antibiotic treatment regimen, both systemic and with a drug-eluting cement spacer (prereimplantation). Patient records were reviewed electronically for preexisting comorbidities, overall health status, synovial fluid cultures, inflammatory serologies, revision surgeries, and recurrent or persistent infection based on the modified Musculoskeletal Infection Society criteria. Patient demographics, ∆CRP, ∆WBC, and time interval to reimplantation were statistically analyzed using receiver operator curves (ROC), Pearson’s correlation coefficient, Levene’s test, and Student’s t-test. Results: Infection-free patients exhibited higher mean CRP and WBC than did patients who were reinfected at both time points. When comparing preexplantation with prereimplantation values, the median ∆CRP was 9.48 mg/L (interquartile range (IQR) 2.3–36.6 mg/L) for patients who did not develop a reinfection versus 2.74 mg/L (IQR 1.4–14.2 mg/L) for patients who developed reinfection (*p* = 0.069). The median ∆WBC was 1.5 × 10^9^/L (IQR 0.6–4.0 × 10^9^/L) for patients who remained infection-free versus 1.2 × 10^9^/L (IQR 0.8–2.2 10^9^/L) for patients who developed reinfection (*p* = 0.072). Analysis of areas under the curve (AUC) using ROC demonstrated poor prediction of persistent infection by ∆CRP (AUC = 0.654) and ∆WBC (AUC = 0.573). Although a highly significant correlation was found between the interim interval and infection persistence (r = 0.655, *p* < 0.01), analysis using ROC failed to result in a specific threshold time to reimplantation above which patients are at significantly higher risk for reinfection (AUC = 0.507). Conclusion: No association could be determined between the delta change in serum CRP and WBC before and after two-stage revision arthroplasty for PJI and reinfection risk. Even though inflammatory serologies demonstrate a downtrending pattern prior to reimplantation, the role of CRP and WBC in determining the optimal timing of reimplantation seems to be dispensable. Planning a second-stage reimplantation requires assessing multiple variables rather than relying on specific numeric changes in these inflammatory marker values.

## 1. Introduction

Periprosthetic joint infections (PJI) are one of the most serious complications following primary hip or knee arthroplasty. Despite advances in diagnostics and treatment, PJI occur in up to 2% of cases and continue to be a source of significant morbidity to patients and an economic burden to society as a whole [1,2,3]. Several therapeutic strategies have been proposed, supporting the ideology that successful eradication is required for the restoration of proper joint functionality [4,5]. Along with antibiotic suppression, treatment entails surgical revision varying from debridement, irrigation, and implant retention (DAIR) [6,7,8] to one-stage exchange [9,10,11] and two-stage revision with temporary placement of antibiotic-loaded cement spacer [12,13,14,15].

With 80–100% success rates [16,17], two-stage revision has been adopted as the gold standard for the treatment of PJI, which entails stage I: explantation of the implant, irrigation and debridement, and implantation of antibiotic-loaded cement spacer, and stage II: explantation of spacer and reimplantation of new implant following extended intravenous and oral organism-specific antibiotic suppression [5,13,14]. Although a time- or parameter-specific algorithm for successful infection eradication has not yet been established, evaluation of clinical presentation, serologic and synovial inflammatory markers, and culture results help guide optimal timing of reimplantation [18,19,20,21].

C-reactive protein (CRP) and white blood cell count (WBC) are two commonly used serum inflammatory markers for assessing infection eradication before reimplantation. Although several studies [22,23,24,25,26,27] have investigated the predictive capabilities of serum inflammatory markers, a clear consensus is yet to exist-while many dismiss their diagnostic efficiency [21,22,23,24,26,27], some support their use as diagnostic assessors of infection status in the interim period before reimplantation [25,28,29]. Previous attempts at defining a specific numeric threshold and significant percentage change of markers, such as CRP and erythrocyte sedimentation rate (ESR), were not successful [22,23,24].

Due to these diagnostic ambiguities, this matched-pair analysis study was carried out with the objective of (1) assessing the role of patient characteristics to the change of serum CRP and WBC values when planning reimplantation in two-stage revision arthroplasty, and (2) investigating the diagnostic practicality of using the change, or delta (Δ), of these serum markers from preexplantation to prereimplantation as an indicator of infection eradication and optimal timing for reimplantation.

## 2. Materials and Methods

The current study is a matched-pair analysis with a chart data review of two-stage revision arthroplasty cases of chronic periprosthetic knee and hip infections at one tertiary referral center.

The diagnosis of PJI prior to the two-stage revision arthroplasty has been based on clinical presentation, serum inflammatory markers (CRP and WBC), synovial fluid analysis, and intraoperative cultures following the American Academy of Orthopaedic Surgeons (AAOS) Clinical Practice Guidelines [5]. The two-stage revision consisted of stage I: explantation of the implant and (if used) bone cement, irrigation and debridement, and implantation of a static antibiotic-loaded cement spacer (polymethylmethacrylate (PMMA)) in knees or an articulating spacer in hips, along with systemic organism-specific antibiotic treatment for 2 weeks intravenously and for an additional 4 weeks orally, and stage II: explantation of the spacer and reimplantation of a new implant following a full 2-week antibiotic holiday period, absence of clinical signs of infection, and negative repeat joint aspiration.

Patient demographics including age, gender, body mass index (BMI), operative joint (hip or knee), health status according to the American Society of Anesthesiologists (ASA) classification score, clinical presentation of infection, comorbidities, synovial fluid analysis, causative organisms cultured at the time of the explantation and their respective antibiograms, time intervals between explantation and reimplantation of prosthesis, number and types of prior revision surgeries, hospitalization time, and serum inflammatory markers’ (CRP and WBC) values throughout the whole inpatient treatment were recorded.

The primary database query yielded 440 potential subjects who underwent a first-time two-stage revision treatment for chronic periprosthetic knee or hip infection in our institution. Among those, 263 subjects with acute infection (<3 weeks of symptoms), infection due to inflammatory arthropathy (rheumatoid arthritis, ankylosing spondylitis, psoriatic arthritis, and systemic lupus erythematous), fungal causative species, or preexisting osteosynthetic material were excluded. Furthermore, we excluded cases with incomplete data, particularly referred cases lacking preexplantation inflammatory markers. We were left with 177 cases of first-time two-stage revision arthroplasty for chronic PJI for investigation (Figure 1). Sample size analysis conducted using the G * Power program with a type I error rate of alpha of 0.05, a power of 85%, and effect size of 0.3 yielded a total sample of 75 subjects.

The remainder of the potential cases for investigation were assembled by propensity score matching in regard to BMI, age, and ASA classification score, and the final sample consisted of 78 subjects: 39 patients with a reinfection (criteria see below) and 39 controls of total hip (THA) and total knee arthroplasties (TKA) who met the modified Musculoskeletal Infection Society (MSIS) diagnostic criteria for PJI [30], with a follow-up time of 29 months and complete serology reports throughout the whole treatment protocol.

Blood samples for testing were obtained for serum CRP (mg/L) and WBC (10^9^/L) at two selected time points: preexplantation, prior to implant removal, and prereimplantation, prior to implant reimplantation. Each blood sample was sent to our Department of Laboratory Medicine for the determination of CRP turbidimetrically using a Cobas c System (Roche Diagnostics International AG, Basel, Switzerland) and WBC using a Sysmex XN (Sysmex Corpotation, Kobe, Hyogo, Japan) instrument. These are the routine procedures for determination of CRP and WBC in our hospital. Reliability and validity measures were reported high for these serum markers (intraclass correlation coefficient > 0.9%). The thresholds for the upper limits of normal of 6 mg/L for CRP and 10 × 10^9^/L for WBC were applied as references. Infection-related failures were determined using the evidence-based and validated updated MSIS diagnostic criteria for PJI [30].

Descriptive statistics including patient demographics were analyzed. Pearson’s correlation coefficient (r) was used to determine the relationship between patient characteristics (age, gender, BMI, ASA classification score, and comorbidities) and status of infection as well as serum inflammatory markers (CRP and WBC) and interim interval to reimplantation. Using Levene’s test for equality of variances and Student’s t-test, changes (∆) in CRP and WBC were investigated. These serum inflammatory markers’ values were reported as medians, means with 95% confidence intervals (95% CI), interquartile ranges (IQR) and standard deviation (SD). Heterogeneity was assessed using box plots. Receiver operator characteristic (ROC) testing was used for the area under the curve (AUC) regarding defining a threshold value for changes in CRP and WBC and interim interval period. A *p*-value of <0.05 was considered to be statistically significant. All statistical analyses were performed using IBM SPSS software version 23 (IBM Corporation, Armonk, NY, USA).

## 3. Results

The data on 78 patients (27 (34.6%) males and 51 (65.4%) females) with a median age of 67 years were meticulously analyzed. As expected, no statistical significance was observed between sex, age, BMI and ASA grade of the patients (Table 1), as these parameters were used for the propensity score matching of the study cohort. Although they were statistically significant (*p* < 0.05), weak correlations (r < 0.5) were observed between patients’ ASA grade, comorbidities (diabetes mellitus type 2, rheumatoid arthritis, chronic kidney disease, and chronic obstructive pulmonary disease (COPD)) and serum inflammatory markers (Table 1). Furthermore, patients who remained infection-free were found to have more comorbid conditions (Figure 2).

Patients who remained infection-free had higher mean CRP and WBC than patients who became reinfected in both the preexplantation (45.4 versus 18 mg/L and 8.6 versus 7.5 × 10^9^/L, respectively) and prereimplantation (11.1 versus 2.8 mg/L and 6.6 versus 6.5 × 10^9^/L, respectively) time points (Table 2). Even though equal variances were assumed (*p* < 0.05) when using Levene’s test for equality of variances, Student’s t-test demonstrated no statistically significant difference (*p* = 0.071) in the mean WBC values of reinfection and infection-free groups. Furthermore, the mean CRP values of reinfection and infection-free groups were found to be not statistically significantly different (*p* = 0.69).

Moreover, a highly heterogeneous distribution of the markers in both selected time points was noted in the box plots (Figure 3 and Figure 4), with outliers particularly among the CRP.

The mean interim interval between explantation and reimplantation in the entire study cohort was 13.7 weeks. Mean time to reimplantation was 13.6 (SD ± 4.7) weeks in the group of patients who remained infection-free compared to 13.8 (SD ± 5.6) weeks in the group of patients with persistent infection. Although a highly significant correlation was found between the interim interval and infection persistence (r = 0.655, *p* < 0.01), no significant correlation was found between interim interval and ∆CRP (r = 0.207, *p* = 0.071) or ∆WBC (r = 0.103, *p* = 0.374) with regard to infection status. Analysis using the ROC curve failed to highlight a specific threshold time to reimplantation above which patients are at significantly higher risk for reinfection (AUC = 0.507) (Figure 5).

As represented in Table 3, the median ∆CRP between preexplantation and prereimplantation was 9.48 mg/L (IQR, 2.3–36.6 mg/L) for patients who did not develop a reinfection versus 2.74 mg/L (IQR, 1.4–14.2 mg/L) for patients who became reinfected (*p* = 0.069). The median ∆WBC was 1.5 10^9^/L (IQR, 0.6–4.0 10^9^/L) for patients who remained infection-free versus 1.2 10^9^/L (IQR, 0.8–2.2 10^9^/L) for patients who developed a reinfection (*p* = 0.072). Furthermore, analysis of AUCs using ROC demonstrated poor prediction of infection status by ∆CRP (AUC = 0.654) and ∆WBC (AUC = 0.573) (Table 3, Figure 6).

## 4. Discussion

∆CRP and ∆WBC are inadequate diagnostic assessors of infection persistence when planning reimplantation in two-stage revision arthroplasty. Although similar observations [21,22,23,24,26,27] have been documented for CRP, no study we are aware of has investigated if the change seen in serum WBC might serve as a more accurate marker. Despite the fact that these markers have been found to be useful, and their use in the clinical practice might be practical [25,28,29,31], the proper diagnosis of persistent PJI relies on a combination of factors and modalities, such as patients’ comorbidities and characteristics which were found to be weakly associated with changes in serum CRP and WBC and strongly associated with incidence of recurrent infection, rather than a single change in value. Although a strong correlation was found between the interim interval and infection persistence, ∆CRP and ∆WBC did not provide any guidance to the optimal timing of reimplantation.

Even though serum CRP and WBC demonstrated a downward trend before reimplantation (Table 2), ∆CRP and ∆WBC were found to be poor predictors of persistent infection. Similar findings have been published by Stambough et al. [24] regarding establishing a percentage change of CRP indicative of infection persistence prior to reimplantation. It has been noted however, that since two-stage revision arthroplasty is considered to be a relatively successful course of action, future studies investigating values of successfully treated patients with those who became reinfected require a very large sample size to reach adequate power [24]. Investigating larger patient cohorts might be of interest since the *p*-values in our ROC analysis (*p* = 0.069 and 0.072, Table 3) were fairly close to 0.05 but failed to establish statistically significant ∆CRP and ∆WBC cutoff values. Nonetheless, false negative or low values of inflammatory serologies could occur in the context of suppressive antibacterial treatment, low-virulence pathogens or chronic PJI as in our investigated cases [32].

While CRP and WBC values may be homogeneous prior to explantation, they can demonstrate a highly heterogeneous distribution of values and outliers at both the preexplantation and prereimplantation time points, as observed in Graph 3 and 4. Furthermore, change in the values of these markers may be somewhat sporadic, erratic, and unpredictable, as previously documented [22] and as exhibited in Table 2, which demonstrates that patients who remain infection-free had higher mean CRP and WBC than patients who become reinfected. Even though inflammatory serologies might be dependent on the causative organism, as documented by Ryu et al. [33] who found that increased preexplantation CRP values in a methicillin-resistant *Staphylococcus aureus* infection were associated with an increased risk of reinfection and longer treatment duration needed for CRP to normalize, our study did not investigate individual causative species. The effect of the various virulent organisms on the change of inflammatory markers prior to reimplantation should be investigated in future studies.

Downtrending inflammatory serologies create a false sense of security. The widely used strategy of simply relying on serum inflammatory markers until they return to normal values is based on limited evidence and may possibly further aggravate the infection, leading to higher inter-stage morbidity and failure rates [34]. Although our data demonstrated declining CRP and WBC values prior to reimplantation regardless of infection persistence, similar to previous documentations [22,23,24], analysis could not reveal any significant differences in these values that could help guide reimplantation. A possible justification might be the small number of investigated cases in our study, as better statistical assessment might be achieved using a larger patient cohort. Nevertheless, no other study of which we are aware of has investigated the trends in WBC values between the two stages of two-stage revision arthroplasty.

While previous reports [34,35] investigated the optimal timing for reimplantation, a clear consensus has yet to be made. Analysis of our data demonstrated a statistically significant, strong correlation between the time to reimplantation and infection persistence, even though the time to reimplantation was almost identical in both groups (Table 2). Nevertheless, attempts at defining a specific threshold time to reimplantation using ROC statistical analysis were not successful (Figure 5). Previous investigation [34] has documented similar findings and found that delaying the time to reimplantation did not significantly improve treatment outcome. Furthermore, although antibiotic-loaded cement spacers are used for their antimicrobial capabilities, they may also develop bacterial biofilm [36,37], making the interim period highly relevant to treatment success. Surgeons might be guided by declining inflammatory marker values throughout the spacer retention period; nevertheless, regression analysis demonstrated that ∆CRP and ∆WBC could not sufficiently determine optimal timing for reimplantation. One of the methods proposed, and applied in our study, for determining infection eradication prior to reimplantation is repeat joint aspiration for obtaining synovial microbial cultures [18,27,38]. Although this technique has shown promising results in reducing the incidence of recurrent infection following reimplantation [18], it has been associated with sampling errors, high false-negative rates and poor sensitivity [22,38,39]. Despite these conflicting documentations that emphasize the tremendous complexity of assessing recurrent infection prior to reimplantation, our institution no longer supports repeat joint aspiration prior to reimplantation, due to its poor sensitivity and prediction of causative species [38,39,40].

Several studies have documented that preexisting conditions such as diabetes mellitus, obesity, smoking, rheumatoid arthritis, and cardiac disease may predispose patients to reinfection or complicate the healing process and ability to eradicate infection [41,42,43]. Analysis yielded weak, yet statistically significant, correlations between patients’ ASA grade, comorbidities and inflammatory markers’ values in both selected time points (Table 1). Although the logical assumption might be that patients with preexisting comorbidities and poor general health may be more prone to infection persistence, descriptive statistics demonstrated that patients who were found to have more comorbidities remained infection-free (Figure 2). Furthermore, no significant association to infection persistence was observed among cases of poor host conditions such as diabetes mellitus, contrary to previous studies [42,43]. However, diabetic patients with higher ASA score exhibited statistically significant, weak correlations to CRP values at both time points. This finding might be explained by natural elevation of inflammatory markers in an immunosuppressive state [44]. Although previous investigation [28] reported moderate sensitivities and high specificities of CRP to detect persistent infection in patients with inflammatory arthritis, our analysis failed to exhibit any significant association between infection persistence and CRP in rheumatic patients. Nevertheless, statistically significant correlations were found between infection persistence and WBC at both time points in this patient group (Table 1). These controversial findings question the usefulness of inflammatory marker values, that have been previously documented to be inaccurate and affected by immunosuppressive therapy [45,46], for the diagnosis of persistent infection in the setting of inflammatory arthropathies. A thorough review of the patient’s risk factors and comorbidities is required when investigating the risk of infection persistence. Among others, renal and pulmonary functions are known to be important in the host’s ability to respond to the treatment of musculoskeletal infection [47,48]. Of the many comorbidities investigated, only patients with COPD had statistically significant correlation to infection persistence following reimplantation (Table 1), even though no significant association was found between COPD patients and inflammatory markers in both time points. These findings are supported by a previous study [49] which reported more prevalent treatment failures of two-stage revision arthroplasty for fungal PJI in patients with renal disease and COPD. Interestingly enough, significant associations between patients with chronic kidney disease and CRP and WBC values were only observed prior to explantation (Table 1). One proposed explanation for this finding might be increased CRP values through diminished filtration in end-stage renal disease, as previously reported [50].

Even though this matched-pair analysis investigated a well-established treatment protocol used in our institution for a long period, it is not without limitations. First, this study was limited by its retrospective nature; data regarding systemic host factors and limb status, both of which have been recognized by the MSIS classification system for their role in influencing treatment outcome [51], were not captured. Additionally, neither individual causative species nor their effects on the changes in inflammatory markers were investigated, due to the extreme difficulty of statistically analyzing a great variety of pathogens. This might be essential, since resistant bacteria and difficult-to-treat organisms have been reported [33] to significantly influence inflammatory serologies. Furthermore, data regarding the type of antibiotic-loaded cement spacer (articulating versus static) used was not acquired. Second, even though repeat synovial fluid analysis was assessed to classify positive results as reinfection, synovial fluid cultures have been documented [52] to have varying sensitivities ranging from 45% to 75%, the misclassification of patients as persistently infected or infection-free might have been possible. Third, information regarding prolonged oral antibiotic suppressive treatment following reimplantation was not gathered. Since results of testing for an underlying infection under oral antibiotic treatment are more likely to be false negative [22,38,39], cases of reinfection might have been miscategorized. Fourth, the patient population is inhomogeneous, so that THA and TKA were included; however, as mentioned above, it is extremely difficult to generate adequate case numbers of reinfections after two-stage exchange assuming a treatment success rate of more than 90% with two-stage treatment. Last, the patient population was still treated according to an old treatment principle (repeated aspirations, drug holidays). In the meantime, we have further developed the treatment principle and largely follow the recommendations of Trampuz et al. [53]. We are looking forward to publish the results of the new treatment principles with a minimum follow-up of 3 years.

In conclusion, no association could be determined between the delta change in serum CRP and WBC between both stages of two-stage revision arthroplasty for PJI and reinfection risk. Even though inflammatory serologies demonstrate a downtrending pattern prior to reimplantation, the role of CRP and WBC in determining the optimal timing of reimplantation seems to be dispensable. Planning a second-stage reimplantation requires assessing multiple factors and modalities (the immunological status of the patient, the local soft-tissue and bony characteristics, and the microbiological profile) rather than relying on a specific numeric change of inflammatory marker values.

## Figures and Tables

**Figure 1 antibiotics-11-01174-f001:**
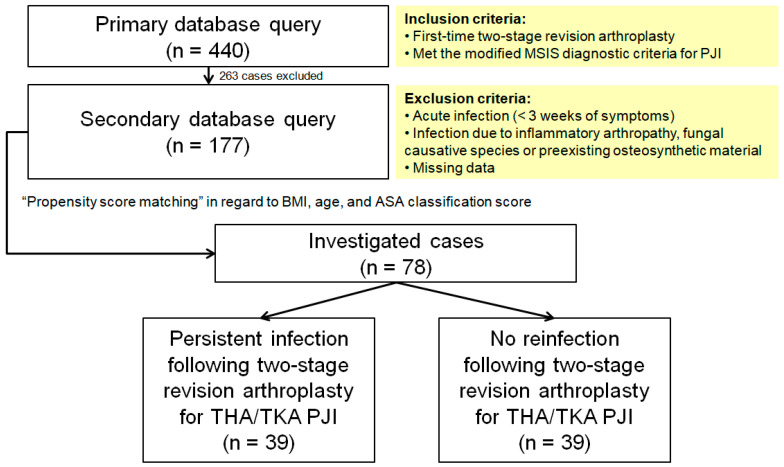
Flow diagram presenting patient selection into the study cohort. MSIS, Musculoskeletal Infection Society; PJI, periprosthetic joint infection; BMI, body mass index; ASA, American Society of Anesthesiologists; THA, total hip arthroplasty; TKA, total knee arthroplasty.

**Figure 2 antibiotics-11-01174-f002:**
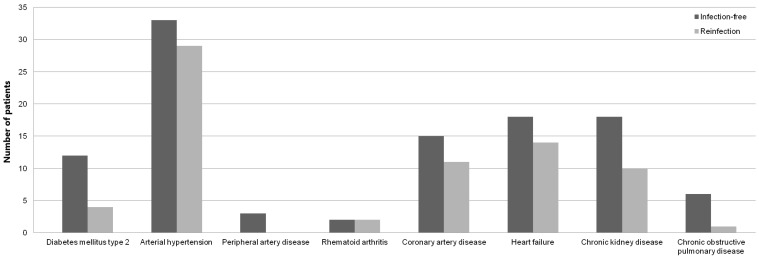
Bar diagram showing number of infection-free and reinfected patients’ comorbidities.

**Figure 3 antibiotics-11-01174-f003:**
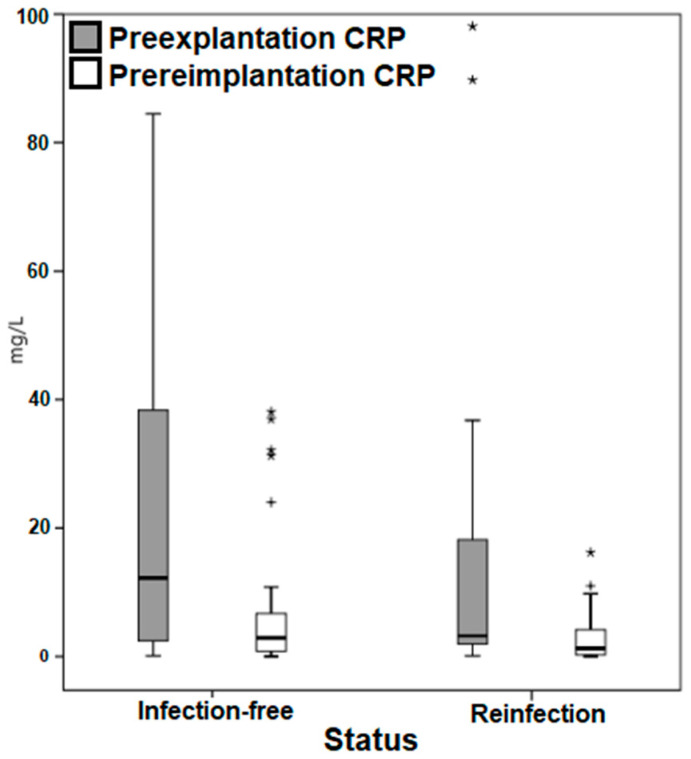
Box plot demonstrating the distribution of the C-reactive protein values. CRP, C-reactive protein. * Outliers. ^+^ Outliers.

**Figure 4 antibiotics-11-01174-f004:**
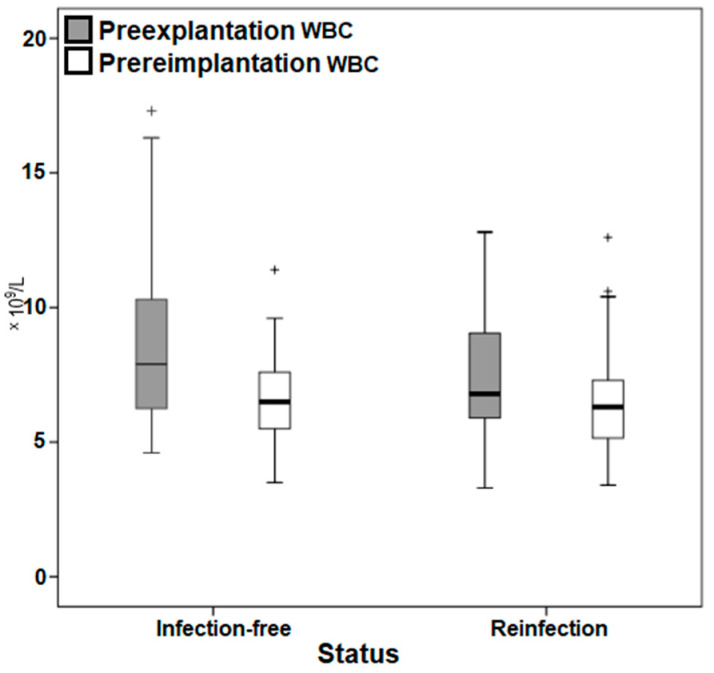
Box plot demonstrating the distribution of the white blood cell count values. WBC, white blood cell count. ^+^ Outliers.

**Figure 5 antibiotics-11-01174-f005:**
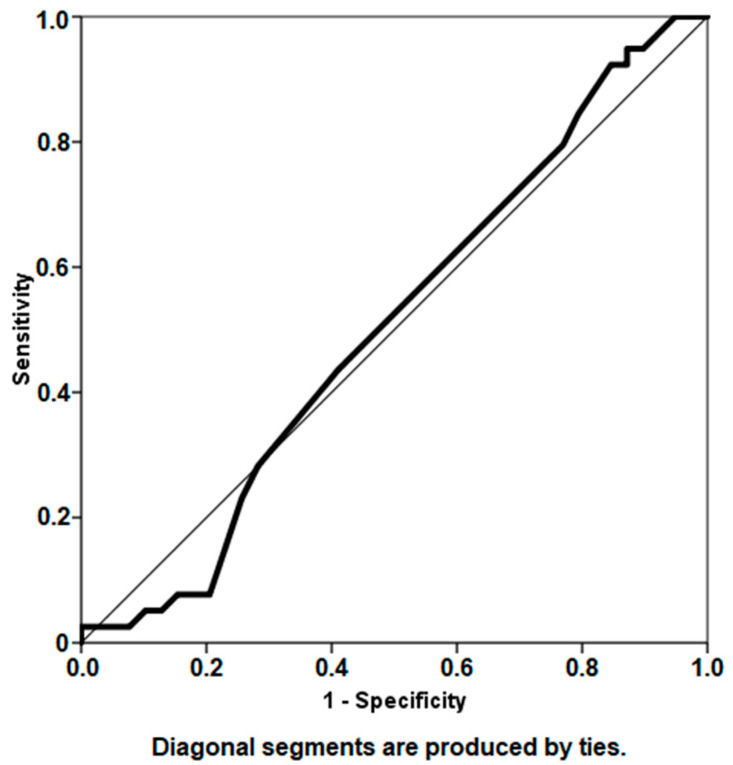
Receiver operator characteristic curve analysis depicting the interim interval period as a predictor for infection persistence (area under curve 0.507).

**Figure 6 antibiotics-11-01174-f006:**
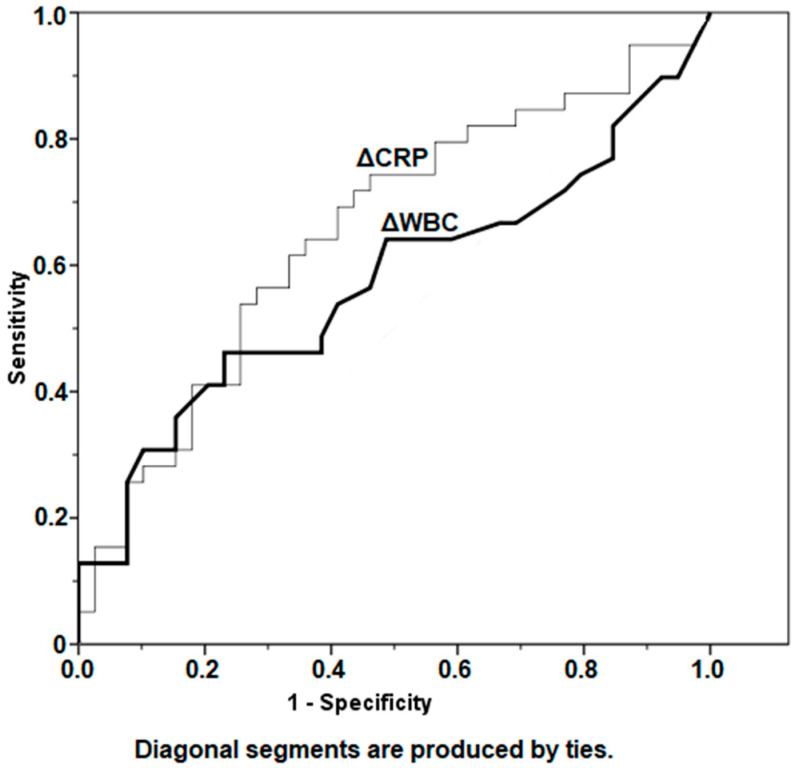
Depiction of change (∆) of CRP and WBC values between the two selected time points: preexplantation and prereimplantation, using receiver operator curves. CRP, C-reactive protein; WBC, white blood cell count.

**Table 1 antibiotics-11-01174-t001:** Pearson’s correlation coefficients for patient characteristics and status of infection as well as serum inflammatory markers at the two selected time points.

		Patient Characteristics											
		Sex	Age	BMI	ASA Grade	DM	AH	PAD	RA	CAD	HF	CKD	COPD
**Infection status**		−0.135	0.047	−0.125	−0.019	−0.217	−0.127	−0.200	0.000	−0.109	−0.104	−0.186	−0.224
**CRP**	Preexplantation	0.173	0.081	−0.036	0.307	0.274	0.091	0.074	0.216	0.006	0.005	0.318	0.056
	Prereimplantation	0.102	0.063	0.035	0.295	0.241	−0.152	0.070	0.116	−0.062	−0.051	0.160	0.129
**WBC**	Preexplantation	0.185	0.046	0.120	0.197	0.123	0.182	0.110	0.293	0.138	0.146	0.270	0.130
	Prereimplantation	−0.025	0.009	−0.038	−0.037	−0.091	0.033	−0.064	0.374	0.089	0.056	−0.046	0.131

Values are highlighted with regard to their statistical significance. Grey, no statistical significance since these parameters were used for “propensity score matching” of the study cohort; Light red, correlation is significant at the 0.05 level; Dark red, correlation is significant at the 0.01 level. BMI, body mass index; ASA, American Society of Anesthesiologists; DM, Diabetes mellitus type 2; AH, arterial hypertension; PAD, peripheral artery disease; RA, rheumatoid arthritis; CAD, coronary artery disease; HF, heart failure; CKD, chronic kidney disease; COPD, chronic obstructive pulmonary disease.

**Table 2 antibiotics-11-01174-t002:** Mean, minimum, maximum and standard deviation of CRP and WBC values and time interval between the two selected time points for patients who were reinfected versus those who remained infection-free.

Parameter	Status	Time Point	Mean	Minimum	Maximum	Standard Deviation
**CRP (mg/L)**	Infection-free	Preexplantation	45.4	0.1	298.4	74.8
		Prereimplantation	11.1	0.01	171	28.3
	Reinfection	Preexplantation	18	0.1	202.3	37.3
		Prereimplantation	2.8	0.01	16.2	3.7
**WBC (10^9^/L)**	Infection-free	Preexplantation	8.6	5	17	10.1
		Prereimplantation	6.6	3.5	11.4	2.9
	Reinfection	Preexplantation	7.5	3	13	5.4
		Prereimplantation	6.5	3.4	12.6	3.5
	Interim interval (weeks)		13.7	4	35	5.3
	Infection-free		13.6	5	35	4.7
	Reinfection		13.8	4	33	5.6

CRP, C-reactive protein; WBC, white blood cell count.

**Table 3 antibiotics-11-01174-t003:** Changes in serum CRP and WBC with regard to the infection status.

Parameter	Status	Median	IQR for Median	Mean	95% CI for Mean	AUC	*p*-Value
**ΔCRP (mg/L)**	Infection-free	9.48	2.3–36.6	38.1	16.5–59.7	0.654	0.069
	Reinfection	2.74	1.4–14.2	15.5	3.6–27.4		
**ΔWBC (10^9^/L)**	Infection-free	1.5	0.6–4.0	2.5	1.7–3.3	0.573	0.072
	Reinfection	1.2	0.8–2.2	1.7	1.2–2.2		

CRP, C-reactive protein; WBC, white blood cell count; IQR, interquartile range; 95% CI, 95% confidence interval; AUC, area under curve.

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
