# Peer review of "Against the Norm: Do Not Rely on Serum C-Reactive Protein and White Blood Cell Count Only When Assessing Eradication of Periprosthetic Joint Infection"

_antibiotics, 2022, doi:10.3390/antibiotics11091174_

Round 1
Reviewer 1 Report
In the current article by Khury et al., the authors present a study on data recorded from patients who underwent two-stage revision arthroplasty for the treatment of periprosthetic joint infections. The authors performed retrospective analyses to investigate the reliability of serum CRP and WBC changes as marker for infection eradication, and hence as parameters to determine the timing of reimplantation. The authors suggest making a more comprehensive assessment of the patient’s condition, including comorbidities to determine the reimplantation schedule, instead of focusing on the serum CRP and/or WBC labels. While the conclusions are based on data from a small cohort and need further validation, they are reasonable and should be of interest to readers/clinicians. It is appreciated that the authors have also discussed some of the limitations of their current study. There are some comments to be addressed that would benefit the article.
1. The authors present the statistical analysis of their matched patient demographics in Table 1. However, it is recommended that they provide some details of the demographics in their methods.
2. The labeling of the graphs (e.g., axes, units, legends) need to be adjusted/enlarged for better legibility.
Author Response
- Reviewer 1 comment:
In the current article by Khury et al., the authors present a study on data recorded from patients who underwent two-stage revision arthroplasty for the treatment of periprosthetic joint infections. The authors performed retrospective analyses to investigate the reliability of serum CRP and WBC changes as marker for infection eradication, and hence as parameters to determine the timing of reimplantation. The authors suggest making a more comprehensive assessment of the patient’s condition, including comorbidities to determine the reimplantation schedule, instead of focusing on the serum CRP and/or WBC labels. While the conclusions are based on data from a small cohort and need further validation, they are reasonable and should be of interest to readers/clinicians. It is appreciated that the authors have also discussed some of the limitations of their current study. There are some comments to be addressed that would benefit the article
- The authors present the statistical analysis of their matched patient demographics in Table 1. However, it is recommended that they provide some details of the demographics in their methods.
Author response:
Thank you very much for these comments and the thorough analysis of our manuscript.
Author action:
With regard to the above mentioned comment, we have provided details of the demographics in the Results section (Lines 151-152).

Reviewer 2 Report
1. Please state how the sample size was achieved.
2. Please adhere to the concerned EQUATOR guidelines and provide a checklist as a supplementary file.
3. State the laboratory method used for estimation of WBC and CRP in detail with details of appropriate validity and reliability measures.
Author Response
- Notes from Reviewer 2:
- Reviewer 2 comment:
- Please state how the sample size was achieved.
Author response:
Thank you very much for this valuable suggestion.
Author action:
Details regarding sample size analysis were added to the Materials and Methods section (Lines 110-112).
- Reviewer 2 comment:
- Please adhere to the concerned EQUATOR guidelines and provide a checklist as a supplementary file.
Author response:
Thank you for this helpful comment.
Author action:
A checklist of the EQUATOR STARD 2015 guidelines will be provided as a supplementary file.
- Reviewer 2 comment:
- State the laboratory method used for estimation of WBC and CRP in detail with details of appropriate validity and reliability measures.
Author response:
We thank you for these important remarks.
Author action:
The laboratory methods used for the estimation of the serum markers including measures of validity and reliability were explained in the Materials and Methods section (Lines 123-131).
